# Private Doctors’ Perspective towards “Patient First” in TB Diagnostic Cascade, Hisar, India

**DOI:** 10.3390/diagnostics14111164

**Published:** 2024-05-31

**Authors:** Sanjeev Saini, Banuru Muralidhara Prasad, Ajay Mahajan, Akshay Duhan, Anuj Jangra, Jitendra Gauttam, Mandeep Malik, Jyoti Kayesth, Bhavin Vadera, Reeti Desai Hobson

**Affiliations:** 1Infectious Disease Detection and Surveillance (IDDS), ICF, New Delhi 110037, India; sanjeevdr1980@gmail.com (S.S.);; 2State Indian Medical Association, Hisar 125001, Haryana, India; 3Directorate of Health Services, Government of Haryana, Hisar 125001, Haryana, India; 4Pandit Bhagwat Dayal Sharma Post Graduate Institute of Medical Sciences, Rohtak 124001, Haryana, India; 5Joint Effort for Elimination of Tuberculosis, JEET, Hisar 125001, Haryana, India; 6United States Agency for International Development, New Delhi 110021, India; 7Infectious Disease Detection and Surveillance (IDDS), ICF, Rockville, MD 20850, USA

**Keywords:** private doctors, private laboratory, tuberculosis, diagnostic algorithm, one-stop model

## Abstract

TB diagnosis has been simplified in India following advances in available diagnostic tools. This facilitates private doctors’ “patient first” approach toward early diagnosis; however, costs remain high. India’s NTEP established a TB diagnostic network, which is free for patients and incentivizes private doctors to participate. Drawing from this context led to the design and implementation of the One-Stop TB Diagnostic Solution model, which was conducted in the Hisar district, Haryana, allowing specimens from presumptive TB patients from private doctors to be collected and tested as per NTEPs diagnostic algorithm. A subset of data pertaining to private doctors was analyzed for the project period. Qualitative data were also collected by interviewing doctors using a snowball method to capture doctors’ perception about the model. Out of 1159 specimens collected from 60 facilities, MTB was detected in 32% and rifampicin resistance was detected in 7% specimens. All specimens went through the diagnostic algorithm. Thirty doctors interviewed were satisfied with the services offered and were appreciative of the program that implements this “patient centric” model. Results from implementation indicate the need to strengthen private diagnostics through a certification process to ensure provision of quality TB diagnostic services.

## 1. Background

Globally, private doctors have a “patient first” approach in their routine tuberculosis (TB) care and treatment practices. During routine TB care and treatment services, doctors conduct patient examination, request diagnostic tests, and prescribe treatment regimens. The diagnosis can be completed in a single consultation or through multiple consultations. To attain appropriate and timely treatment regimens, doctors require access to relevant diagnostic tools and awareness of the approved diagnostic algorithms available from the National TB Elimination Program (NTEP). As the newer diagnostic tools come into the market, nearly 70% of decisions are made by doctors based on laboratory results [1]. In India, the private laboratory service market equipped with TB diagnostic tests has quality-assured diagnostic services certified by the National Accreditation Board for Testing and Calibration Laboratories (NABL) or the College of American Pathologists (CAPs). Laboratories performing TB sputum and chest X-rays are more readily available in urban areas, whereas those performing liquid culture tests are available in metro-cities. Therefore, doctors are dependent on clinical presentation of patients or chest X-rays to screen for TB. Often, doctors resort to treatment in lieu of complete diagnosis of TB in the best interest of patients [2]. Based on clinical progress, patients are referred to medical colleges or research institutions by primary care doctors.

TB diagnosis follows a systematic approach or diagnostic algorithm, where the results of preceding tests determine the requirement of further tests. Until recently, availability and accessibility to newer TB diagnostic services had been a challenge in India. With global advocacy and renewed commitment from countries to meet the United Nations High-Level Meeting (UNHLM) for TB 2023 goals, investments are increasing in the TB diagnostic arena [3]. In India, recent advancements were spurred by the launch of GeneXpert machines, a rapid molecular assay that has enabled accurate and rapid diagnosis of TB, providing results comparable to the gold standard for TB diagnosis—the liquid-based culture. To further improve the quality of the TB diagnostic network, standards of TB care in India (STCI) were launched in 2014 and widely disseminated to support private doctors in TB patient diagnosis and treatment [4]. The Central TB Division (CTD) led efforts to engage private doctors as part of the TB diagnostic network using STCI working through partners, TB advocates, medical colleges, and collaborative institutions. The NTEP also focused on partnerships with private healthcare providers with strategies directed at encouragement, and unconditional and conditional incentives. These were funded jointly through India’s domestic budget and through external donor funded projects, mainly the Global Fund, such as Joint Efforts for Elimination of TB (JEET) and Project Axshya. In addition, the NTEP set up systems to support presumptive and diagnosed TB patients from the private sector to access rapid molecular tests such as GeneXpert and Truenat, in public facilities and free of cost [5]. However, the ongoing systemic issues within the public health facilities remain a challenge for meaningful engagement with private doctors. These include (a) limited working hours, (b) availability of laboratory technicians, (c) huge workloads (specimens flowing from public health facilities), and (d) limited availability of test cartridges [6,7].

The experiences of engaging private laboratories highlighted the need to design and implement a private laboratory engagement model that provided an end-to-end diagnostic solution for presumptive TB patients (PTBPs) in both public and private facilities. Under the leadership of the CTD and the Infectious Disease Detection and Surveillance (IDDS) project funded by the United States Agency for International Development (USAID), the “One-Stop TB Diagnostic Solution” was implemented in the Hisar district of Haryana in collaboration with the State TB Cell and District TB Office. The model design, implementation process, and results were published recently by Rajesh R. et al., (2023) [8]. This current paper focuses on experiences and perspectives of the private doctors who were engaged through this model and how their “patient first” approach to TB care was affected.

### Concerning the Model Strategy and Implementation Geography

The One-Stop TB Diagnostic Solution model is a “patient centric” approach that offers an end-to-end diagnostic solution by engaging a NTEP-certified private laboratory to provide services as per the program’s diagnostic algorithm. Private doctors in Hisar, through their clinical acumen, identify PTBPs or diagnosed TB patients and guide them to provide quality specimens for further confirmation. The specimens collected are subjected to an upfront nucleic acid amplification test (NAAT), first- and second-line line-probe assay (FL-LPA and SL-LPA), and liquid culture and drug susceptibility testing (LC-DST) [9]. Prior to conducting the tests, the PTBPs are registered by the respective private doctor’s facility in the NTEP’s TB management information system web portal (Ni-kshay). Patient results are updated as and when available against the identity of the patients in the portal. The doctors can view the test results immediately via their mobile phone through a Ni-kshay application. IDDS project staff also shared the results via email or short message systems to support timely treatment decision-making.

The Hisar district of Haryana has an approximate population of two million with an estimated TB prevalence of 465 (326–605) persons per 100,000 population [10]. Prior to the model, specimen collection from private hospitals and individual clinics occurred ad hoc, funded through the NTEP budget and the district Rotary Club. The approach resulted in more than 50% of total district notifications coming from the engaged private hospitals and doctors. However, as part of the One-Stop model, the engaged NTEP-certified private laboratory (Thyrocare, Mumbai, India) established a systematic process of specimen collection using runners who collected specimens as and when they became available from public and private facilities [8]. The IDDS team provided continued medical education (CME) sessions in the district to raise awareness on the model and the updated diagnostic algorithm. These meetings reinforced that the doctor/s at their respective facilities would collect the specimen directly and inform the runners, versus referring PTBPs to district hospitals or higher-level TB centers. Each of the specimens collected had duly filled-in test request forms (TRF) with a Ni-kshay ID of the PTBPs/patient attached. The engaged private laboratory, Thyrocare, then processed specimens as per the NTEP TB diagnostic algorithm at their Gurugram and Mumbai, laboratories. Results were entered into the Ni-kshay portal matching the IDs available in the TRF to support patient notification and initiation of treatment.

## 2. Methods

### 2.1. Data Source

This study included both quantitative and qualitative data sources. The quantitative data were collected using a project recording and reporting system. The qualitative data were used to capture the perspectives from doctors and collected through an open-ended questionnaire.

Quantitative data (subset data) pertaining only to specimens collected from private doctors were analyzed for the period 16 May 2022 to 31 January 2023. The data were verified using the Ni-kshay portal by district NTEP staff and the implementing IDDS team. Reporting variables such as the date of specimen collection, type of specimen collected, type of patients (PTBP or diagnosed TB patient), diagnostic tests performed, and issuance of results were also collected.

Qualitative data were collected using a pre-tested, open-ended questionnaire using a snowball approach, which was administered randomly to doctors who had referred at least one PTBP or a diagnosed TB patient to the model. Hard copies of the questionnaire were shared with doctors to provide their perspective on TB diagnostic services. The duly filled-in questionnaires were collected from January 2024 to February 2024. The runners collected the filled-in questionnaires and submitted them to the IDDS team.

### 2.2. Analysis and Statistics

The data analysis was performed using SPSS 16.0 software. All specimens with valid results generated through the model were considered as output variables. For the outcome analysis (as outlined in Rajest R., et al. [8]), we considered specimens with the following characteristics:Xpert negative results as MTB-negative and no further tests were conducted.Xpert positive results (MTB detected and rifampicin-sensitive results) and first-line LPA—rifampicin- and isoniazid-sensitive results were drug-susceptible TB (DS-TB).Xpert positive results (MTB detected and rifampicin-resistant results) and with first line LPA—rifampicin- and/or isoniazid-resistant results; second line LPA—showing resistant to fluoroquinolones; and liquid culture DST results showing resistance to any of the following drugs: moxifloxacin, linezolid, and pyrazinamide (drug-resistant TB (DR-TB)).

The analysis is presented in the Results as per tests performed. All specimens were tracked from the point of collection to the issuance of results as mentioned above for analysis of turnaround time (TAT). Qualitative data collected was analyzed manually and codes were applied to group responses to ensure anonymity. Analysis focused on the knowledge of diagnostic services, availability of these services in the district, perception of services offered through the model, and recommendations to the NTEP on refining and scaling-up the model.

The model was implemented in collaboration with NTEP under routine programmatic conditions. Data were provided by the program and through the project management information system. However, the interviews with doctors were performed through a questionnaire, and doctors’ written consent was sought for the information collected. Since the model was implemented through the program, we did not seek any formal ethical approvals.

## 3. Results

### 3.1. Profile of Patients from Private Doctors

A total of 1159 patient specimens collected from 60 private doctors/facilities were processed during the project period. The number of specimens increased from 36 to 98, reaching a peak of 180 specimens in September (Figure 1). During the project period, 13 facilities continued to provide specimens monthly. Over 10,000 specimens were collected in the model: 9924 pulmonary and 240 extra pulmonary. Out of the total extra pulmonary specimens, 50% were from private doctors (Table 1). The private doctors received specimens from patients mostly in urban areas (86%), male patients (66%), and from patients above 30 years (80%) of age.

### 3.2. Patients Completing the Diagnostic Algorithm

Upfront GeneXpert (Xpert MTB/RIF) tests were performed for 1159 specimens that were collected (Table 2). Of the specimens collected, MTB was detected in 32% of the specimens and rifampicin-resistant (RR) was detected in 7% of the specimens. The 374 specimens with MTB-positive results were also processed by LPA, LC, and LC DST.

Out of the total 374 processed for smear test prior to FL-LPA, 315 specimens showed smear-positive results. The 59 smear-negative specimens were processed for LC; 13 specimens showed growth. One insufficient sample was not processed. Overall, a total of 327 specimens were processed for FL-LPA tests for one of the first-line drugs (rifampicin/isoniazid). The results showed 291 specimens sensitive for both first-line drugs. From the test results, isoniazid mono resistance was detected in 14 specimens (Figure 2). Specimens that showed resistance to any one of the first-line drugs were processed by SL-LPA. The results of SL-LPA showed that six specimens were resistant to fluoroquinolones (FQ).

The smear-negative specimens and specimens that showed resistance in the SL-LPA tests were processed for LC as per the NTEP algorithm. Out of 92 specimens which were process for liquid culture; growth was observed in 29 specimens (13 specimens prior to LPA). One specimen was contaminated. The drug susceptibility test (DST) results for 28 specimens were available and 5 specimens showed resistance to pyrazinamide and one specimen showed resistance to moxifloxacin. One specimen was resistant to both moxifloxacin and pyrazinamide. A total of 349 specimens with rifampicin-sensitive (RS) TB and 25 specimens with rifampicin-resistant (RR) TB, completed the diagnostic algorithm (see Appendix A).

Specimen test results were issued within the stipulated TAT for respective tests along the diagnostic-care cascade. The median number of days for reporting results in overall model and for specimens from private doctors was similar for Xpert, FL-LPA, SL-LPA, and LC-DST, which was 1, 5, 7, and 45 days, respectively.

### 3.3. Private Doctors’ Perspectives concerning the Model

Fifty percent of private doctors who participated in the One-Stop model provided a written response to the open-ended questionnaire. The results of analysis are grouped into (a) knowledge of diagnostic services and availability of these services in the district, (b) perception about the model services, and (c) recommendations for the NTEP. Based on the responses, doctors’ knowledge of TB tests is broadly segregated by screening tests, confirmatory tests, and other tests. Almost all 30 doctors used the following screening tests routinely in their practice: (a) chest X-ray (CXR) and (b) Mantoux tests. About 50% (i.e., 15 doctors) also reported to have used interferon gamma release assay (IGRA) to screen for TB among patients with clinical symptoms. More than 70% of the doctors responded that they utilized an acid-fast bacilli (AFB) test followed by Xpert to confirm TB diagnosis. However, when specimens were not available, 37% of the doctors resorted to radiological tests (high-resolution computed tomography—HRCT) along with clinical presentation. Most of these tests were available in the district and Xpert tests were available at the district TB hospital. Other tests such as erythrocyte sedimentation rate (ESR) (83%) and complete blood count (CBC) (60%) were also used as part of the TB screening tests. The following response is from a doctor explaining their use of CXR.

“Chest X-rays are easily available, low cost to patients, and help in screening for tuberculosis in routine practice”doctor

All the private doctors were satisfied with the services offered through the model. They believed that the model facilitated early diagnosis of TB patients without delaying specimen collection and transportation and availability of test results. Following the results, those with DR-TB were referred to the district TB hospital for management. The One-Stop model also enhanced learning of the NTEP TB diagnostic algorithm. The following are views from doctors regarding the model.

“Yes, I’m aware of the One-Stop TB Diagnostic Solution. I think it is a good solution for the early diagnosis of TB without delay in transportation. Poor patients can avail diagnostic services free of cost.”doctor

“Yes, all services and reporting’s are good. Staff do their job with responsibility and good care of patients”doctor

“Yes, the One-Stop TB Diagnostic Solution model has been a major change into the healthcare sector as it provides us with quick interpretation and better understanding of clinical outcomes.”doctor

“Yes, the One-Stop TB Diagnostic Solution model has helped us understand the TB diagnostics in a more robust manner”doctor

Private doctors appreciated the efforts of the district TB program and administration in implementing the innovative model that has facilitated ensuring early diagnosis of TB. Among the key recommendations to program, there is a growing demand to implement the One-Stop model on a continuum basis in the district, and strengthen current TB diagnostic services with prompt receipt of test results, especially for Xpert tests. The improved turnaround times have enabled doctors to initiate appropriate treatment for their patients.

“Yes, it was good and needs to be implemented again”doctor

“Yes, good efforts on the part of the government sector”doctor

## 4. Discussion

The NTEP has been supporting private doctors through initiatives related to specimen collection, honoring referrals from private doctors for universal drug susceptibility tests (UDST), and providing patient-wise drug boxes for treatment management. However, the “One-Stop TB Diagnostic Solution” model is unique as the approach empowered private doctors to take appropriate treatment decisions as they received test results within stipulated time and in line with the NTEP’s diagnostic algorithm. All 1159 specimens that were referred to the model followed the algorithm and the results were available to doctors via email and or through the Ni-kshay application.

Currently, models to engage private doctors in diagnostic services are limited. One recent example is of the “DOST” intervention model, where 9331 specimens from private doctors/facilities were transported by an interface agency to Xpert sites at public facilities between July 2019 and December 2020 [11]. In this model, only 68% of the specimens completed the LPA tests. In contrast, the One-Stop model had ~99% specimens processed according to the diagnostic algorithm. This could be the result of engagement of the NTEP-certified private laboratory to perform the tests.

Private doctors in the Hisar district routinely use CXR and the Mantoux test to screen TB patients. Doctors were also found to have used additional tests, such as CBC and ESR, which are readily available in the district and are affordable to patients. In another study from Karnataka, the 235 private providers interviewed prescribed CXR (87%) and other tests like ESR (33%), Mantoux (40%), Xpert (25%), and culture tests (about 5%) for diagnosis of TB [12]. A study from Mumbai also highlighted the private doctors’ preference for CXR when compared to doctors in public health facilities [13]. Other than CXR and Mantoux tests, use of CBC or ESR to screen for TB is yet to be determined, although they may be used in conjunction with other NTEP-approved diagnostic tests [14].

Pulmonologists and general medicine practitioners expressed the need to learn about the DR-TB status of patients, and this need was satisfied through the One-Stop model. They also appreciated the results of LPA, liquid culture, and DST results more specifically for extra pulmonary TB. In the model, all 143 EPTB specimens were processed for Xpert and liquid culture, and >90% had microbiological results. This was possible because a private laboratory was engaged. In a study from central India, 1220 EPTB patients were interviewed and results showed 83% of patients did not receive microbiological confirmation and 66% received empirical treatment [15]. Empirical treatment in the absence of confirmative diagnosis is evident from a study among private doctors in Patna and Mumbai [16]. However, the current study is limited to the diagnostic-care cascade and we did not consider the treatment management of patients. In the process of ensuring prompt diagnosis, the model has imparted knowledge of the NTEP diagnostic algorithm and adherence to STCI. According to the available literature, adherence to STCI among private doctors was limited [12,17,18]. Another study from Jharkhand, India, where 17 private practitioners were interviewed, also highlighted the need for CME to ensure complete diagnostics, access to diagnostic tools through innovative partnership with governments, and robust supply-chain management within the public health system [19].

The One-Stop model was designed to support the PTBP specimen journey through completion of the diagnostic-care cascade, irrespective of whether they were collected at a public or private facility. Private doctors were sensitized to the model through CME sessions where there was an opportunity to reinforce knowledge of the NTEP’s TB diagnostic cascade. However limited emphasis was placed on continued referrals to the model. Secondly, doctors continued to refer patients directly to the nearest public health facility for sputum microscopy or to the district TB hospital for Xpert tests. These specimens were considered as specimens from public sector and the tests results were reported accordingly. The One-Stop model provided all necessary test results in compliance with the NTEP diagnostic algorithm; most doctors referred DR-TB patients to the district TB hospital for management. The limitations of model is discussed as part of the “One-Stop TB Diagnostic Solution” model, where the reasons for specimens not completing diagnostic algorithm are elaborated [8]. In addition to those mentioned, the current study questionnaire focused on services and was limited to capture the doctors’ knowledge about TB tests and or the diagnostic algorithm. Overall, inferences from the study highlight that the model is instrumental in facilitating clinical decision-making by private providers toward positive patient-level outcomes.

## 5. Conclusions

The patient-centric approach underpinning the One-Stop model is unique in that it demonstrated private doctors’ willingness to participate in the NTEP to expedite TB patient diagnosis during routine practice. The model provided results for tests that are outlined in the NTEP’s diagnostic algorithm within stipulated TAT, which supported these doctors in making appropriate treatment decisions in a timely manner. Given the high rate of patients accessing private services, results from the model highlight the need to continue to strengthen and improve the quality of diagnostic services in the private sector. This could be possible through a process of certification for private sector diagnostic laboratory services and monitored by NTEP.

## Figures and Tables

**Figure 1 diagnostics-14-01164-f001:**
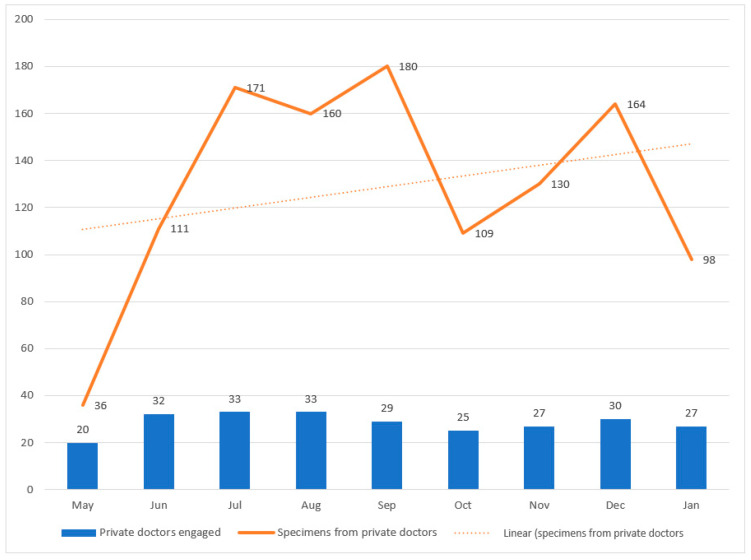
Month-wise trend in the number of private doctors providing specimens during the project period.

**Figure 2 diagnostics-14-01164-f002:**
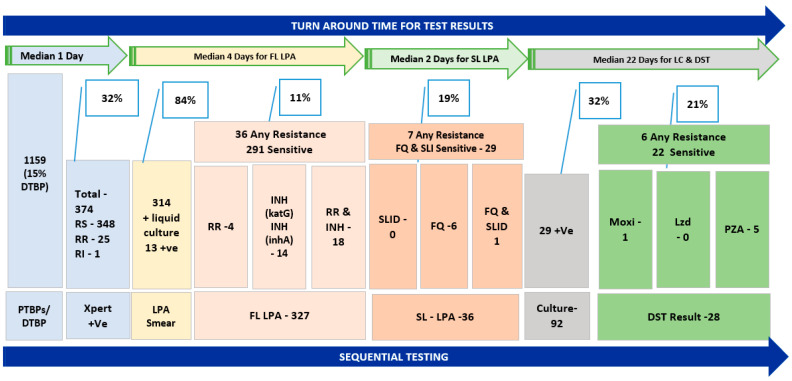
Specimens from private doctors completing TB diagnostic tests as per NTEP algorithm and median turnaround time under the One-Stop TB Diagnostic Solution model. Abbreviations: PTBPs—presumptive TB patients, DTBP—diagnosed TB patients, RS—rifampicin-sensitive, RR—rifampicin-resistance, INH—isoniazid, LPA—line probe assay, FL-LPA—first-line LPA, SL-LPA—second-line LPA, SLI—second-line injectables, SLID—second-line injectable drugs, FQ—fluoroquinolones, +ve—positive, Moxi—moxifloxacin, Lzd—linezolid, PZA—pyrazinamide, LC—liquid culture, LC & DST—liquid culture and drug susceptibility testing, DST—drug susceptibility tests.

**Table 1 diagnostics-14-01164-t001:** Demographic and clinical profile of patients’ specimens collected in the model.

Variable	Overall, from Model (Number)	Private Doctors (Number)	Private Doctors (Percentage)
Total Specimens in the Model	10,164		
Private Doctors Type of Patient		1159	11%
PTBPs	9468	985	85%
Diagnosed TB Patients	696	174	15%
Type of Specimen			
Pulmonary	9924	1016	88%
Extra Pulmonary	240	143	12%
Geography			
Rural	6660	168	14%
Urban	3504	991	86%
Gender			
Male	6663	769	66%
Female	3500	390	34%
Age (years)			
≤14	284	18	2%
15–29	2249	206	18%
30–44	2309	227	20%
45–59	2411	293	25%
≥60	2911	415	36%

**Table 2 diagnostics-14-01164-t002:** Xpert result for specimens collected from private doctors.

Xpert Result for MTB	Model Total (n)	Private Doctors (n)	Private Doctors (%)
Total	10164	1159	
MTB Detected	2152	374	32%
MTB Not Detected	8012	785	68%
Rifampicin Status Xpert			
Rif-Resistant	134	25	7%
Rif-Sensitive	1996	348	93%
Rif-Indeterminate	22	1	

Rif—rifampicin.

## Data Availability

The data pertaining to the model are analyzed and presented in the manuscript. The data are part of the routine program and reported in the Ni-kshay portal. This may be available upon request to concerned officials of the National TB Elimination Program.

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
