# Peer review of "Private Doctors’ Perspective towards “Patient First” in TB Diagnostic Cascade, Hisar, India"

_diagnostics, 2024, doi:10.3390/diagnostics14111164_

Round 1
Reviewer 1 Report
Comments and Suggestions for Authors
Dear Authors
This is an interesting article with great practical utility as it describes an approach to diagnosing tuberculosis in India involving private doctors.
The participation of private doctors in anti-Tb actions is recognized as difficult and necessary. In some countries, all actions related to TB are developed in the public system, thus dispensing medicines and epidemiological notification records allow for easier information across the country.
The article had a qualitative and quantitative approach and presented successful results. Thus, half of private doctors were satisfied with the "patients first" approach.
My suggestions are: a brief contextualization of epidemiological data from India to understand the situation in Hisar and report on some aspects of the 60 private doctors interviewed. For example: what would be the average sex, age of doctors; time since graduation etc.
Author Response
Many thanks for your kind words. The programmes continued efforts have paved way for the increase participation of private doctors in TB care cascade.

Reviewer 2 Report
Comments and Suggestions for Authors
This article highlights the pressing issue of the significant gap between the number of tuberculosis (TB) patients in India and the availability of doctors for diagnosis and treatment. The proposed solution of utilizing certified private diagnosis labs presents a promising approach to address this challenge effectively.
The author advocates for a patient-centric approach, proposing a one-stop model where private doctors collaborate with the National Tuberculosis Elimination Program (NTEP) to streamline TB patient diagnosis. By leveraging the expertise and resources of private diagnosis labs, doctors can expedite the diagnostic process, leading to faster treatment decisions and improved patient outcomes.
Overall, the article presents a compelling argument for adopting this model, emphasizing its potential to enhance the efficiency of TB diagnosis and treatment in India.
Comments on the Quality of English LanguageMinor editing of English language required.
Author Response
Many thanks for the kind words. As you have rightly pointed the emphasis is on engaging the existing private laboratory through a process of certification to strength the diagnostic care cascade in India. The programme has taken measures to review the laboratory capacities across India and is also encouraging private laboratories to certify the key diagnostic tests.